# FineCLIPER: Multi-modal Fine-grained CLIP for Dynamic Facial Expression Recognition with AdaptERs

## Abstract

Dynamic Facial Expression Recognition (DFER) is crucial for understanding human behavior. However, current methods exhibit limited performance mainly due to the scarcity of high-quality data, the insufficient utilization of facial dynamics, and the ambiguity of expression semantics, etc. To this end, we propose a novel framework, named Multi-modal Fine-grained CLIP for Dynamic Facial Expression Recognition with AdaptERs (**FineCLIPER**), incorporating the following novel designs: 1) To better distinguish between similar facial expressions, we extend the class labels to textual descriptions from both positive and negative aspects, and obtain supervision by calculating the cross-modal similarity based on the CLIP model; 2) Our FineCLIPER adopts a hierarchical manner to effectively mine useful cues from DFE videos. Specifically, besides directly embedding video frames as input (*low semantic level*), we propose to extract the face segmentation masks and landmarks based on each frame (*middle semantic level*) and utilize the Multi-modal Large Language Model (MLLM) to further generate detailed descriptions of facial changes across frames with designed prompts (*high semantic level*). Additionally, we also adopt Parameter-Efficient Fine-Tuning (PEFT) to enable efficient adaptation of large pre-trained models (*i.e.*, CLIP) for this task. Our FineCLIPER achieves SOTA performance on the DFEW, FERV39k, and MAFW datasets in both supervised and zero-shot settings with few tunable parameters. Analysis and ablation studies further validate its effectiveness. Code and dataset will be released upon the paper notification.

## CCS Concepts

• **Computing methodologies → Computer vision**; • **Human-centered computing → Human computer interaction (HCI)**.

## Keywords

Dynamic Facial Expression Recognition, Multi-Modal, Model Adaptation, Parameter-Efficient Transfer Learning, Contrastive Learning

## 1 Introduction

Facial expressions are important signals to convey human emotions, thus accurately recognizing them has significant meaning

*ACM MM, 2024, Melbourne, Australia*
© 2024 Copyright held by the owner/author(s). Publication rights licensed to ACM.
ACM ISBN 978-x-xxxx-xxxx-x/YY/MM
https://doi.org/10.1145/nnnnnnn.nnnnnnn

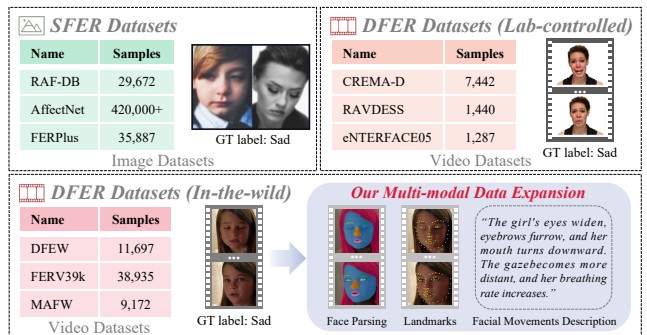

**Figure 1: Visualization of dataset example and quantity.**

for various tasks, including interpersonal communication, human-computer interaction (HCI) [16, 22, 36], mental health diagnosing [2, 21, 47], driving safety monitoring [56, 62, 63], etc. Traditional Facial Expression Recognition (FER) resorts to static images. However, since dynamic emotional changes could not be well-represented within a single image, research attention has been shifted to Dynamic Facial Expression Recognition (DFER), which distinguishes the temporally displayed facial expressions in videos.

The study of DFER algorithms starts from highly-controlled environments [3, 37, 42], where faces are frontal and non-blurry [30, 59] as shown in the upper right of Fig. 1. However, such an ideal assumption makes the obtained models vulnerable to real-world situations. Therefore, researchers have turned to more open scenes and constructed several *in-the-wild* DFER datasets, *e.g.*, DFEW [18], FERV39k [60], and MAFW [33], to facilitate the development of corresponding methods [27, 28, 35, 39, 57, 68]. While category labels are treated without semantic meanings (*e.g.*, *Happiness* may only be represented by a class id *"0"*), recent research on DFER [11, 26, 52, 70] has further delved into the exploration of vision-language multi-modal learning beyond the traditional classification paradigm based on CNNs [10, 20, 24, 50], RNNs [1, 61], and transformers [25, 69].

Although huge efforts have been spent, the performance of DFER methods still suffers from noisy frames, small inter-class differences, and ambiguity between expressions, making it inappropriate to adopt video/action recognition techniques directly. Specifically, to distinctly improve the performance of DEFR algorithms, we have to face the following unique and tough challenges: 1) The ambiguity of semantic labels for dynamic facial expressions, and 2) The subtle and nuanced movements of local face parts (*i.e.*, skeletons, muscles, etc.). The first challenge originates in the difficulty of accurate human labeling and complex expression ways adopted by different persons, while the latter demands additional focus on fine-grained details that happen in specific regions within a human face.

To tackle the above challenges, we propose a novel framework called FineCLIPER, short for Multi-modal Fine-grained CLIP for

Dynamic Facial Expression Recognition with AdaptERs. Specifically, we utilize the Contrastive Language-Image Pretraining (CLIP) model [46], which is particularly suitable for providing a cross-modal latent space. To avoid the huge cost of fine-tuning such large pre-trained models, FineCLIPER adopts the Parameter-Efficient Fine-Tuning (PEFT) strategy by adding several adaption modules with small parameters for tuning (as shown in Fig. 2), achieving high efficiency while preserving the remarkable performance.

Specifically, our FineCLIPER has the following characteristics that distinguish it from previous works:

Firstly, by adopting the vision-text learning paradigm, we transform the ground truth label to form the textual supervision (*e.g.*, *"A person with an expression of {Label}"*). But one noteworthy innovation is that we meanwhile generate and use the negative counterparts (*e.g.*, *"A person with an expression of **No** {Label}"*). Such label augmentation via PN (Positive-Negative) descriptors is inspired by the negative prompting strategy [7, 43], and found to be useful here for differentiating between ambiguous categories. A notable progress is observed in the "Disgust" category of the DFEW dataset, while most baselines [4, 25, 29, 34, 57] suffer from a nearby 0% accuracy, our FineCLIPER significantly promotes the performance by more than 25%, as shown in Tab. 2.

Furthermore, we adopt a *semantically hierarchical strategy* to comprehensively mine useful information from the input video data. Specifically, features from directly embedding video frames stand at a relatively *low* semantic level. For *middle* semantic level, we utilize a well-trained face analysis model (*i.e.* FaceXFormer [44]) to extract the face segmentation masks and landmarks from each frame. Intuitively, the former offers prior about face structures while the latter provides specific pivots for model attention. Additionally, we try to obtain descriptions at a *high* semantic level for describing dynamic facial changes across frames. This is realized by leveraging a well-trained MLLM, Video-LLaVA [31] to act as a facial expression analyst following given template-based prompts, and the generated descriptions will be carefully refined. All the above features at various semantic levels will be integrated to obtain the final representation of a given video.

To summarize, our contributions are as follows:

- We introduce FineCLIPER, a novel multi-modal framework that enhances Dynamic Facial Expression Recognition (DFER) through extensively mining useful information at different semantic levels from the video data, and all the obtained features (*i.e.*, features embedded from visual frames, face segmentation, face landmarks, and the extra fine-grained descriptions obtained via MLLM) are integrated finally to serve as a more comprehensive overall representation;
- To address the ambiguity between categories, we propose a label augmentation strategy, not only transforming the class label to textual supervision but also using a combination of both positive and negative descriptors;
- Extensive experiments conducted on DFER datasets, *i.e.*, DFEW, FERV39k, and MAFW, show that our FineCLIPER framework achieves new state-of-the-art performance on both supervised and zero-shot settings with only a small number of tunable parameters. Comprehensive ablations and analyses further validate the effectiveness of FineCLIPER.

## 2 Related Work

**Dynamic Facial Expression Recognition.** In early DFER research, the focus was on developing diverse local descriptors on lab-controlled datasets [3, 37, 38]. Then the rise of deep learning and accessible in-the-wild DFER datasets [18, 33, 60] leads to new trends towards DFER research. The first trend [10, 20, 24] involves the direct use of 3D CNNs [12, 54, 55] to extract joint spatio-temporal features from raw videos. The second trend [9, 19, 50, 61] combines 2D CNNs [5, 48] with RNNs [5, 13] for feature extraction and sequence modeling. The third emerging trend integrates transformer [8], as demonstrated in works like Former-DFER [69], STT [40], and IAL [25]. These methods combine convolutional and attention-based approaches to enhance the understanding of visual data, especially in distinguishing samples based on varying visual dynamics. However, in prior efforts, the semantic meaning of class labels is neglected, and insufficient attention has been paid to the subtle and nuanced movements of the human face. Therefore, based on the well-trained large cross-modal models (i.e. CLIP), we propose to extend the class label to textual supervision both positively and negatively. Moreover, to fully exploit the visual information within videos, we also design a hierarchical information mining strategy to generate representative video features, which is a weighted fusion of various features involving different semantic levels, including video frame feature, the middle-level facial semantics from segmentation maps and detected landmarks, we well as the high-level semantics encoded from fine-grained descriptions provided by MLLM.

**CLIP in Classification.** Vision-Language Models (VLMs), *e.g.*, CLIP [46], have recently demonstrated superior performance across various tasks, including video understanding [6, 45, 58, 66], 3D generation or editing [15, 17, 32], and region profiling [53, 64], etc. CLIP leverages a vast corpus of image-text pairs to ground its framework in contrastive learning, resulting in robust pre-trained image and text encoders that demonstrate remarkable feature extraction capabilities. Recent studies [26, 52, 70] have also applied CLIP to the DFER task. Among them, $A^3$lign-DFER [52] introduces a comprehensive alignment paradigm for DFER through a complicated design. CLIPER [26] adopts a two-stage training paradigm instead of end-to-end training; however, it is limited in capturing temporal information. Furthermore, DFER-CLIP [70] incorporates a transformer-based module to better capture temporal information in videos, but it requires fully fine-tune the image encoder and the proposed temporal module during training, leading to inefficiency.

However, while these works have explored the semantic information of labels compared to traditional DFER, they often overlook the interrelations among facial expressions and the individual differences among humans as they directly extend labels into relevant action descriptions (*e.g.*, Happiness→smiling mouth, raised cheeks, wrinkled eyes, ... [70]). This oversight can lead to further ambiguity. In light of this, we propose PN (Positive-Negative) descriptors, extending the ground truth labels from contrastive views to better distinguish between ambiguous categories.

## 3 Methodology

In this section, we first briefly go through the overall pipeline and basic notations of the framework in Sec. 3.1. Then, we elaborate

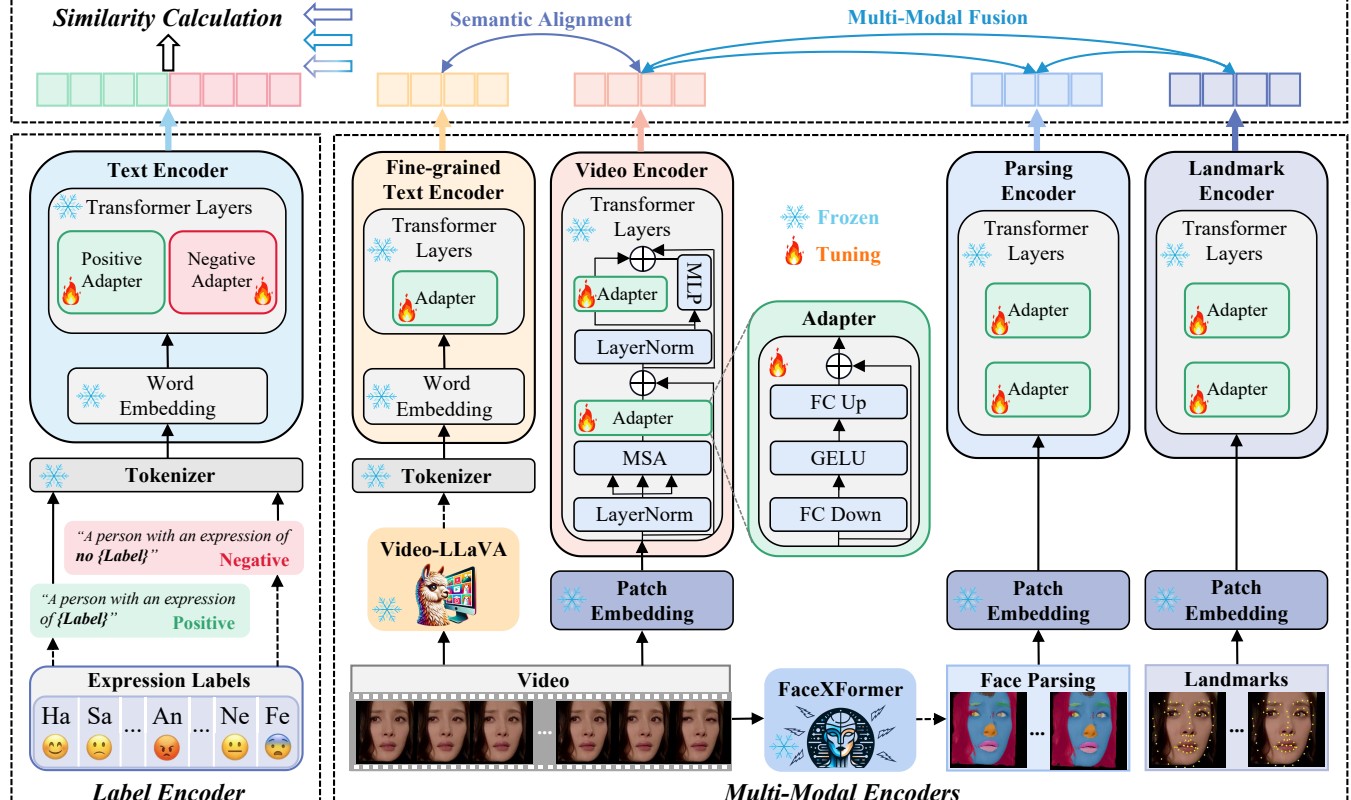

**Figure 2: The FineCLIPER framework can be divided into three main components: Label Encoder, Multi-Modal Encoders, and Similarity Calculation. The Label Encoder augments labels using PN descriptors, followed by PN adaptors within text encoder; The Multi-Modal Encoders handle hierarchical information mined from low semantic levels to high semantic levels of human face; The Similarity Calculation module further integrates and computes the similarities of the representations obtained earlier via contrastive learning.**

on how to augment the original class labels to obtain positive-negative textual supervision in Sec. 3.2, followed by details about our hierarchical information mining strategy to obtain multi-modal features in Sec. 3.3. The integration of diverse features is introduced in Sec. 3.4. The overall pipeline is illustrated in Fig. 2.

### 3.1 Overall Pipeline

Formally, given a video clip $V$, the task of DFER aims to recognize the facial expression label $Cls$. Using text templates as *"A person with an expression of {Cls}"*, the class label could be further transformed into textual supervision, which could better utilize the semantic meaning of the category name.

Let $\mathcal{V}$ represents a set of videos and $C$ denotes collections of augmented textual descriptions of labels, our framework could produce the embedded representations for both a given video and its corresponding textual supervision, resulting in $\mathbf{v_i}$ and $\mathbf{c_i}$. Note that in our cases, $\mathbf{v_i}$ is an integration of features from different semantic levels, namely low-level (video frames), middle-level (face parsing and landmarks), and high-level semantics (fine-grained captions of facial action changes obtained using MLLM). The similarity between $\mathbf{v_i}$ and $\mathbf{c_i}$ is calculated as $sim_i$. To employ the cross-entropy

loss, we calculate the prediction probability over class $cls_i$ as:

$$p(cls_i|\mathbf{v_i}) = \frac{\exp(sim_i/\tau)}{\sum_{i=0}^{N-1}\exp(sim_i/\tau)}, \qquad (1)$$

where $N$ is the number of total classes and $\tau$ represents the temperature parameter of CLIP.

### 3.2 Label Augmentation via PN Descriptors

Although in-the-wild DFER usually comprises limited categories (*e.g.*, 7 in DFEW [18] and FERV39k [60], or 11 in MAFW [33]), the recognition difficulty does not reduce due to the high inter-class ambiguity (as shown in Tab. 2). Therefore, as stated in Sec. 3.1, class labels are transformed into textual supervision for utilizing their semantic meanings.

While existing CLIP-based DFER models [26, 52, 70] mostly focus on enriching the textual descriptions for ground truth labels from a positive view, in this work, we devise a different label augmentation strategy by extending the original class labels from both positive and negative perspectives. Specifically, the Positive-Negative (PN) descriptors are derived as follows: *i.e.*, P(ositive): *"A person with an expression of {Cls}."*, and N(egative): *"A person with an expression*

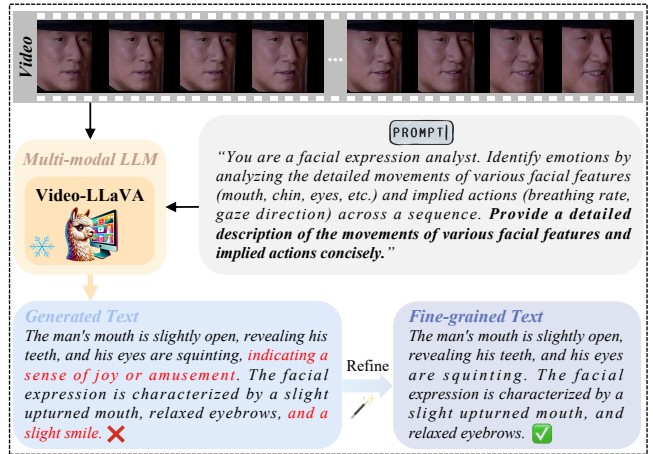

**Figure 3: Fine-grained Text Generation and Refinement.**

of **no** {Cls}.". Correspondingly, the augmented textual supervision $C$ could contain two different collections, namely $C_P$ for positive collections and $C_N$ for negative collections. Then, both text collections are tokenized and projected into word embeddings obtaining $\mathbf{X}_{T_P}, \mathbf{X}_{T_N} \in \mathbb{R}^{l \times d_T}$, where $l$ represents the text length. The inputs are further constructed as:

$$\mathbf{z}_{T_P}^{(0)} = \mathbf{X}_{T_P} + \mathbf{E}_{T_P}, \quad \mathbf{z}_{T_N}^{(0)} = \mathbf{X}_{T_N} + \mathbf{E}_{T_N}, \tag{2}$$

where $\mathbf{E}$ denotes the positional encoding.

To further encode $\mathbf{z}_{T_P}^{(0)}$ and $\mathbf{z}_{T_N}^{(0)}$, we resort to the pre-trained textual part of VLM [46], a model with $L_T$ pre-trained transformer layers, devoted by $\{\mathcal{E}_T^{(i)}\}_{i=1}^{L_T}$. Keeping the original weights of these well-trained layers, we introduce trainable lightweight adapters after each frozen layer $\mathcal{E}_T^{(j)}$. denoted as $\{\mathcal{A}_{T_P}^{(j)}\}$ and $\{\mathcal{A}_{T_N}^{(j)}\}$ for positive and negative textual supervision, respectively. Then the encoded positive and negative textual features could be obtained via:

$$\mathbf{z}_{T_P}^{(j)} = \mathcal{E}_{T_P}^{(j)}(\mathcal{A}_{T_P}^{(j)}(\mathbf{z}_{T_P}^{(j-1)})), \quad \mathbf{z}_{T_N}^{(j)} = \mathcal{E}_{T_N}^{(j)}(\mathcal{A}_{T_N}^{(j)}(\mathbf{z}_{T_N}^{(j-1)})). \tag{3}$$

We adopt the basic Adapter structure proposed in [14] for all adapters in our FineCLIPER framework. The structure of the adapter is illustrated in the middle of Fig. 2. Then the final positive and negative text representations can be obtained by:

$$\mathbf{c}_P = \mathbf{h}_T(\mathbf{z}_{T_P,l}^{(L_T)}), \quad \mathbf{c}_N = \mathbf{h}_T(\mathbf{z}_{T_N,l}^{(L_T)}), \tag{4}$$

where $\mathbf{z}_{T,l}^{(L_T)}$ is the last token of $\mathbf{z}_T^{(L_T)}$ and $\mathbf{h}_T$ is a projection layer.

### 3.3 Hierarchical Information Mining

Our FineCLIPER adopts a hierarchical manner to mine useful information from: 1) *low semantic level*, where video frames are directly embedded; 2) *middle semantic level*, where face segmentation and landmarks are exploited, and 3) *high semantic level*, where fine-grained descriptions are obtained via MLLM to depict facial dynamics across frames. Details can be found as follows:

**Video Frames Embedding** could provide semantically low-level features since the model operates at pixel-level. To effectively explore the spatial-temporal visual information, we resort to the strong spatial modeling abilities displayed by CLIP and utilize a temporal-expanded version inspired by [65].

Formally, given a video clip $V \in \mathbb{R}^{T \times H \times W \times 3}$, where $H \times W$ is the spatial size and $T$ is the temporal length. For $t$-th frame, we spatially divide it into non-overlapping patches $\{\mathbf{P}_{t,i}\}_{i=1}^M \in \mathbb{R}^{P^2 \times 3}$, where $M = HW/P^2$. These patches are then projected into patch embeddings $\mathbf{X}_{v,t} \in \mathbb{R}^{M \times d}$, where $d$ represents the embedding dimension. Therefore, the representation for the given video $V$ could be $\mathbf{z} \in \mathbb{R}^{T \times M \times d}$. After the temporal information undergoes processing by the temporal adapter, the spatially adapted feature can be derived through the following procedure:

$$\mathbf{z}_{TemV}^{(j)} = \mathcal{E}_V^{(j)}(\mathcal{A}_V^{(j)}(\mathbf{z}^{(j)})), \tag{5}$$

$$\mathbf{z}_{SpaV}^{(j)} = \mathcal{E}_V^{(j)}(\mathcal{A}_V^{(j)}(\mathbf{z}_{TemV}^{(j)})), \tag{6}$$

where $\mathbf{z}_{TemV}^{(j)}$ and $\mathbf{z}_{SpaV}^{(j)}$ denotes the temporally and spatially adapted features, respectively.

As a result, the adapter, operating in parallel with the MLP layer, aims to collectively refine the representation of spatiotemporal information. The final feature, scaled by a factor $s$ (set to 0.5 in our framework), can be expressed as follows:

$$\mathbf{z}_V^{(j)} = \mathbf{z}_{SpaV}^{(j)} + MLP(LN(\mathbf{z}_{SpaV}^{(j)})) + s \cdot \mathcal{A}_V^{(j)}(LN(\mathbf{z}_{SpaV}^{(j)})). \tag{7}$$

Thus, the ultimate video representation at a low semantic level is derived as $\mathbf{v} = \mathbf{h}_V(\mathbf{z}_V^{(L_V)})$.

**Face Parsing and Landmarks Detection.** Based on a given frame, we could further mine middle-level semantic information from it. In our task, as the main part of a frame is mostly human faces, we choose to utilize a powerful facial analysis model, FaceXFormer [44], to obtain generalized and robust face representations. Specifically, we extract the facial segmentation map and perform landmark detection. Intuitively, the former implies the semantically grouped facial regions, while the latter could provide accurate locations indicating different face parts (**e.g.**, eyes, nose, etc.)

Specifically, given a specific video clip $V$, the extracted parsing results and landmark maps are represented as $P$ and $L$, respectively. Following patch embedding, both $P$ and $L$ are fed into the corresponding segmentation encoder $\mathbf{E}_P$ and landmark encoder $\mathbf{E}_L$, similar to the operation done for the frame data. The encoders $\mathbf{E}_P$ and $\mathbf{E}_L$ share weights to collaboratively capture middle-level face semantics. Finally, the parsing and landmark representations can be obtained as $\mathbf{p} = \mathbf{h}_P(\mathbf{z}_P^{(L_P)})$ and $\mathbf{l} = \mathbf{h}_L(\mathbf{z}_L^{(L_L)})$, and $\mathbf{h}_P$ and $\mathbf{h}_L$ are projection layers for $P$ and $L$, respectively.

**Additional Fine-grained Descriptions.** In this part, we try to achieve fine-grained details describing the facial dynamics across video frames to serve as high-level semantics. Specifically, for each video clip $V$, we adopt Video-LLaVA [31], a MLLM, to generate detailed descriptions under the guidance of an elaborately designed prompt, where the model is asked to play a role as a facial expression analyst to provide details of facial changes, as illustrated in Fig. 3. To elaborate, the provided text prompt raises requirements for the granularity of the descriptions, explicitly specifying movements

involving various local facial regions. However, the generated description may include emotion-related words associated with the label or contain some redundant information. Hence, we thoroughly refined all generated descriptions to achieve a concise and high-quality summary. The refinement works as follows. Initially, we employed a rule-based approach, utilizing pre-configured regular filters to eliminate redundant and irrelevant textual information. Popular text processing tools from the NLTK package were then utilized to remove noise. Subsequently, each data entry will go through manual inspection to filter out abnormal descriptions.

The average number of tokens in our refined descriptions is approximately 35 tokens. However, research [67] demonstrates the actual effective length of CLIP's text encoder is even less than 20 tokens. Hence, to better explore the fine-grained description of facial changes, we adopt the text encoder of Long-CLIP [67] as our fine-grained text encoder $\mathbf{E}_F$, which can support text inputs of up to 248 tokens. The refined fine-grained description, denoted as $F$, is further tokenized and projected into embeddings $\mathbf{X}_F$. Following a procedure similar to the text encoder described in Sec. 3.2, the input is further constructed as $\mathbf{z}_F^{(0)} = \mathbf{X}_F + \mathbf{E}_F$, where $\mathbf{E}_F$ is the positional encoding of $F$. Subsequently, by feeding it into the projector $\mathbf{h}_F$, we could contain the final feature vector of $F$ as: $\mathbf{f} = \mathbf{h}_F(\mathbf{z}_{F,l}^{(L_F)})$.

## 3.4 Weighted Integration.

Through the aforementioned semantically hierarchical information mining process, we obtain: 1) low-level video frame feature $\mathbf{v}$, 2) middle-level face parsing features $\mathbf{p}$ and face landmark features $\mathbf{l}$, and 3) high-level fine-grained description features $\mathbf{f}$. The integration of these features is done using an adaptive fusion strategy.

Specifically, given a specific video $V$, the supervision for the $i^{th}$ class is represented by both the positive $\mathbf{c}_P^i$ and negative $\mathbf{c}_N^i$. Suppose any representation $\mathbf{m} \in \{\mathbf{v}, \mathbf{p}, \mathbf{l}, \mathbf{f}\}$, the similarity between $\mathbf{m}$ and $\mathbf{c}_P$, as well as $\mathbf{m}$ and $\mathbf{c}_N$ is defined by calculating the cosine similarity:

$$sim_{i,\mathbf{m}}^{pos} = \frac{\mathbf{c}_P^i \cdot \mathbf{m}}{\|\mathbf{c}_P^i\| \|\mathbf{m}\|}, \quad sim_{i,\mathbf{m}}^{neg} = \frac{\mathbf{c}_N^i \cdot \mathbf{m}}{\|\mathbf{c}_N^i\| \|\mathbf{m}\|}, \quad (8)$$

and the final similarity is obtained by: $sim_{i,\mathbf{m}} = sim_{i,\mathbf{m}}^{pos} - sim_{i,\mathbf{m}}^{neg}$, which further distinguishes similarity among similar categories

Then, by finding the max similarity across all the categories, we obtain $sim_{\mathbf{v}} = max_{i=0}^{N}(sim_{i,\mathbf{v}})$. Similarly, we could get $sim_{\mathbf{f}}$, $sim_{\mathbf{p}}$, and $sim_{\mathbf{l}}$ following corresponding max-similarity category. Normalizing these similarities, we obtain the weights corresponding to that representation as:

$$w_{\mathbf{m}} = \frac{e^{sim_{\mathbf{m}}}}{e^{sim_{\mathbf{v}}} + e^{sim_{\mathbf{f}}} + e^{sim_{\mathbf{p}}} + e^{sim_{\mathbf{l}}}}. \quad (9)$$

Such weights could be calculated for $\mathbf{p}, \mathbf{l}, \mathbf{f}$ similarly, resulting in the corresponding weights $w_{\mathbf{v}}$, $w_{\mathbf{f}}$, $w_{\mathbf{p}}$, and $w_{\mathbf{l}}$. Then the overall multi-modal representation $\mathbf{v}^{mm}$ of Multi-Modal Encoders can be obtained as follows:

$$\mathbf{v}^{mm} = w_{\mathbf{v}} \cdot \mathbf{v} + w_{\mathbf{f}} \cdot \mathbf{f} + w_{\mathbf{p}} \cdot \mathbf{p} + w_{\mathbf{l}} \cdot \mathbf{l}. \quad (10)$$

where the weights also correspond to the weights of the cross-entropy loss for each modality. Then the overall loss function can

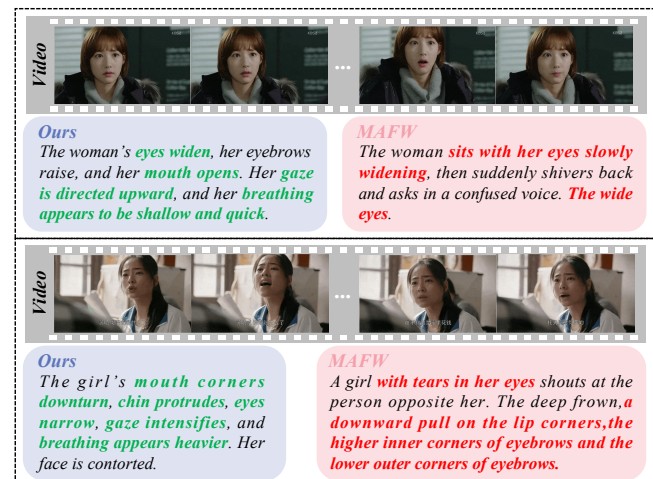

**Figure 4: Comparison of video caption examples between our generated captions and those of the MAFW dataset. Our captions precisely describe facial activities (highlighted in green), in contrast to the MAFW descriptions, which are overly broad and tedious (highlighted in red).**

thus be expressed as:

$$\mathcal{L} = \frac{1}{\mathcal{B}} \sum_{i=1}^{\mathcal{B}} (\mathcal{H}(y_i, p(cls_i|\mathbf{v}^{mm})) +$$
$$w_{\mathbf{v}} \cdot \mathcal{H}(y_i, p(cls_i|\mathbf{v})) + w_{\mathbf{p}} \cdot \mathcal{H}(y_i, p(cls_i|\mathbf{p})) \quad (11)$$
$$+ w_{\mathbf{l}} \cdot \mathcal{H}(y_i, p(cls_i|\mathbf{l})) + w_{\mathbf{f}} \cdot \mathcal{H}(y_i, p(cls_i|\mathbf{f}))).$$

## 4 Experiment

### 4.1 Setup

**Datasets and Evaluation.** Following previous works, we adopt both supervised and zero-shot learning paradigms, evaluating our proposed FineCLIPER together with the baselines on the various in-the-wild DFER datasets, including DFEW [18], FERV39k [60], and MAFW [33]. We utilize UAR (Unweighted Average Recall) and WAR (Weighted Average Recall) as evaluation metrics for our assessments. Both DFEW and FERV39k have 7 dynamic facial expression categories to recognize, while MAFW has 11 categories. It is noteworthy that MAFW dataset comes with video captions for each video, making it a choice for pretraining in zero-shot setting. **Implementation Details.** All the experiments of our FineCLIPER are built on a CLIP model with the backbone of ViT-B/16 using a single NVIDIA RTX 4090 GPU for fairness and consistency. We process the input by resizing and cropping 16 video frames to a uniform size of 224×224 pixels. The SGD optimizer is employed with an initial learning rate of $3 \times 10^{-4}$. FineCLIPER is trained in an end-to-end manner over 30 epochs with the temperature hyper-parameter $\tau = 0.01$.

### 4.2 Main Results

**Supervised Setting.** The quantitative results in the supervised setting on three standard DFER datasets are depicted in Tab. 1. It can

**Table 1: Comparisons of our FineCLIPER with the state-of-the-art Supervised DFER methods on DFEW, FERV39k, and MAFW. \*: FineCLIPER with face parsing and landmarks modalities; †: FineCLIPER with fine-grained text modality. The best results are highlighted in Bold, and the second-best Underlined.**

| Method | Backbone | Tunable Param (M) | DFEW | | FERV39k | | MAFW | |
|---|---|---|---|---|---|---|---|---|
| | | | UAR | WAR | UAR | WAR | UAR | WAR |
| EC-STFL (MM'20) [18] | C3D / P3D | 78 | 45.35 | 56.51 | - | - | - | - |
| Former-DFER (MM'21) [69] | Transformer | 18 | 53.69 | 65.70 | 37.20 | 46.85 | 31.16 | 43.27 |
| CEFLNet (IS'22) [34] | ResNet-18 | 13 | 51.14 | 65.35 | - | - | - | - |
| NR-DFERNet (ArXiv'22) [29] | CNN-Transformer | - | 54.21 | 68.19 | 33.99 | 45.97 | - | - |
| STT (ArXiv'22) [40] | ResNet-18 | - | 54.58 | 66.65 | 37.76 | 48.11 | - | - |
| DPCNet (MM'22) [61] | ResNet-50 (first 5 layers) | - | 57.11 | 66.32 | - | - | - | - |
| T-ESFL (MM'22) [33] | ResNet-Transformer | - | - | - | - | - | 33.28 | 48.18 |
| EST (PR'23) [35] | ResNet-18 | 43 | 53.94 | 65.85 | - | - | - | - |
| Freq-HD (MM'23) [51] | VGG13-LSTM | - | 46.85 | 55.68 | 33.07 | 45.26 | - | - |
| LOGO-Former (ICASSP'23) [41] | ResNet-18 | - | 54.21 | 66.98 | 38.22 | 48.13 | - | - |
| IAL (AAAI'23) [25] | ResNet-18 | 19 | 55.71 | 69.24 | 35.82 | 48.54 | - | - |
| AEN (CVPRW'23) [23] | ResNet-18 | - | 56.66 | 69.37 | 38.18 | 47.88 | - | - |
| M3DFEL (CVPR'23) [57] | ResNet-18-3D | - | 56.10 | 69.25 | 35.94 | 47.67 | - | - |
| MAE-DFER (MM'23) [49] | ViT-B/16 | 85 | 63.41 | 74.43 | 43.12 | 52.07 | 41.62 | 54.31 |
| S2D (ArXiv'23) [4] | ViT-B/16 | 9 | 65.45 | 74.81 | 43.97 | 46.21 | 43.40 | 52.55 |
| CLIPER (ArXiv'23) [26] | CLIP-ViT-B/16 | 88 | 57.56 | 70.84 | 41.23 | 51.34 | - | - |
| DFER-CLIP (BMVC'23) [70] | CLIP-ViT-B/32 | 90 | 59.61 | 71.25 | 41.27 | 51.65 | 39.89 | 52.55 |
| EmoCLIP (FG'24) [11] | CLIP-ViT-B/32 | - | 58.04 | 62.12 | 31.41 | 36.18 | 34.24 | 41.46 |
| A³lign-DFER (ArXiv'24) [52] | CLIP-ViT-L/14 | - | 64.09 | 74.20 | 41.87 | 51.77 | 42.07 | 53.24 |
| FineCLIPER (Ours) | CLIP-ViT-B/16 | 13 | 62.81 | 72.86 | 42.88 | 52.01 | 42.19 | 53.12 |
| FineCLIPER* (Ours) | CLIP-ViT-B/16 | 19 | 64.89 | 75.05 | 44.15 | 52.12 | 43.02 | 54.69 |
| FineCLIPER† (Ours) | CLIP-ViT-B/16 | 14 | 65.72 | 75.01 | 43.86 | 53.02 | 43.91 | 54.11 |
| FineCLIPER*† (Ours) | CLIP-ViT-B/16 | 20 | **65.98** | **76.21** | **45.22** | **53.98** | **45.01** | **56.91** |

**Table 2: Comparative analyses of accuracy across various emotion categories: FineCLIPER vs. other approaches on DFEW.**

| Method | Tunable Param (M) | Accuracy of Each Emotion | | | | | | | DFEW | |
|---|---|---|---|---|---|---|---|---|---|---|
| | | Hap. | Sad. | Neu. | Ang. | Sur. | Dis. | Fea. | UAR | WAR |
| Former-DFER (MM'21) [69] | 18 | 84.05 | 62.57 | 67.52 | 70.03 | 56.43 | 3.45 | 31.78 | 53.69 | 65.70 |
| CEFLNet (IS'22) [34] | 13 | 84.00 | 68.00 | 67.00 | 70.00 | 52.00 | 0.00 | 17.00 | 51.14 | 65.35 |
| NR-DFERNet (ArXiv'22) [29] | - | 88.47 | 64.84 | 70.03 | 75.09 | 61.60 | 0.00 | 19.43 | 54.21 | 68.19 |
| STT (ArXiv'22) [40] | - | 87.36 | 67.90 | 64.97 | 71.24 | 53.10 | 3.49 | 34.04 | 54.58 | 66.65 |
| EST (PR'23) [35] | 43 | 86.87 | 66.58 | 67.18 | 71.84 | 47.53 | 5.52 | 28.49 | 53.43 | 65.85 |
| IAL (AAAI'23) [25] | 19 | 87.95 | 67.21 | 70.10 | 76.06 | 62.22 | 0.00 | 36.44 | 55.71 | 69.24 |
| M3DFEL (CVPR'23) [57] | - | 89.59 | 68.38 | 67.88 | 74.24 | 59.69 | 0.00 | 31.64 | 56.10 | 69.25 |
| S2D (ArXiv'23) [4] | 9 | 93.87 | 83.25 | 75.31 | 84.19 | 64.33 | 0.00 | 37.07 | 62.57 | 75.98 |
| FineCLIPER (Ours) | 13 | 89.99 | 81.79 | 75.42 | 80.12 | 61.03 | 7.12 | 32.98 | 62.81 | 72.86 |
| FineCLIPER* (Ours) | 19 | 92.86 | 83.88 | 76.10 | 83.56 | 64.69 | 13.02 | 38.13 | 64.89 | 75.05 |
| FineCLIPER† (Ours) | 14 | 94.59 | 85.17 | 78.03 | 85.09 | 64.03 | 22.98 | 37.11 | 65.72 | 75.01 |
| FineCLIPER*† (Ours) | 20 | **94.71** | **86.22** | **78.19** | **86.19** | **65.01** | **26.58** | **38.20** | **65.98** | **76.21** |

be observed that our proposed FineCLIPER achieves state-of-the-art performance compared with other DFER approaches. In addition, our method outperforms all CLIP-based DFER methods with the most lightweight architecture and also the least tunable parameters. Furthermore, we investigate three variants of our FineCLIPER, incorporating face parsing and landmark modalities, along with fine-grained text descriptions of facial changes, which justify the combination of these strategies. The superiority of our FineCLIPER is also supported by the substantial improvement in the most challenging category for previous methods, i.e., "Disgust (Dis.)", as

**Table 3: Comparison with state-of-the-art Zero-Shot DFER methods. †: FineCLIPER with fine-grained text modality.**

| Method | Backbone | Pre-training Dataset | DFEW UAR | DFEW WAR | FERV39k UAR | FERV39k WAR | MAFW UAR | MAFW WAR |
|--------|----------|---------------------|-----|-----|-----|-----|-----|-----|
| CLIP (ICML'21) [46] | ViT-B/32 | LAION-400M | 23.34 | 20.07 | 20.99 | 17.09 | 18.42 | 19.16 |
| FaRL (CVPR'22) [71] | ViT-B/16 | LAION Face-20M | 23.14 | 31.54 | 21.67 | 25.65 | 14.18 | 11.78 |
| EmoCLIP (FG'24) [11] | CLIP-ViT-B/32 | MAFW (class description) | 22.85 | 24.96 | 39.35 | 41.60 | 24.12 | 24.74 |
| EmoCLIP (FG'24) [11] | CLIP-ViT-B/32 | MAFW (video caption) | 36.76 | 46.27 | 26.73 | 35.30 | 25.86 | 33.49 |
| FineCLIPER† (Ours) | CLIP-ViT-B/16 | MAFW (video caption) | 47.52 | 57.12 | 34.59 | 42.28 | 34.02 | 40.23 |
| FineCLIPER† (Ours) | CLIP-ViT-B/16 | MAFW (fine-grained caption) | 52.26 | 62.03 | 39.72 | 46.01 | **38.77** | **46.12** |
| FineCLIPER† (Ours) | CLIP-ViT-B/16 | DFEW (fine-grained caption) | **57.48** | **65.45** | 40.10 | 46.91 | - | - |
| FineCLIPER† (Ours) | CLIP-ViT-B/16 | FERV39k (fine-grained caption) | 55.13 | 63.89 | **40.79** | **48.63** | - | - |

**Table 4: Performance of FineCLIPER* w.r.t. data from parsing and landmark modalities on DFEW, FERV39k, and MAFW.**

| Parsing | Land. | DFEW UAR | DFEW WAR | FERV39k UAR | FERV39k WAR | MAFW UAR | MAFW WAR |
|---------|-------|-----|-----|-----|-----|-----|-----|
| ✗ | ✗ | 62.81 | 72.86 | 42.88 | 52.01 | 42.19 | 53.12 |
| ✓ | ✗ | 63.66 | 73.86 | 43.66 | 52.00 | 42.78 | 53.59 |
| ✗ | ✓ | 63.71 | 74.16 | 43.53 | 52.08 | 42.56 | 53.16 |
| ✓ | ✓ | **64.89** | **75.05** | **44.15** | **52.12** | **43.02** | **54.69** |

**Table 5: Performance w.r.t. diverse adapter configurations. *pos* and *neg* are positive and negative adapters, respectively.**

| Text | Video | DFEW UAR | DFEW WAR | FERV39k UAR | FERV39k WAR | MAFW UAR | MAFW WAR |
|------|-------|-----|-----|-----|-----|-----|-----|
| ✗ | ✗ | 59.61 | 71.25 | 41.27 | 51.65 | 39.89 | 52.55 |
| ✓$_{pos}$ | ✗ | 60.32 | 71.55 | 41.51 | 51.70 | 40.47 | 52.62 |
| ✓$_{pos+neg}$ | ✗ | 61.19 | 71.95 | 42.29 | 51.72 | 40.71 | 52.86 |
| ✗ | ✓ | 61.88 | 72.08 | 41.56 | 51.77 | 41.26 | 51.44 |
| ✓$_{pos+neg}$ | ✓ | **62.81** | **72.86** | **42.88** | **52.01** | **42.19** | **53.12** |

shown in Tab. 2. It is worth noting that even without the hierarchical information modeling, FineCLIPER, which only has PN descriptors with adapters, still achieves competitive performance. This demonstrates the effectiveness of the label augmentation strategy via PN descriptors and the usage of PEFT techniques. Further ablation studies can be found in Sec. 4.3.

**Zero-shot Setting.** To assess the generalization ability of FineCLIPER, we perform zero-shot DFER using captions extracted directly from each video. Our main baseline is EmoCLIP [11], which is the first CLIP-based zero-shot DFER model, utilizes the MAFW [33] dataset for pertaining. The comparison between captions in MAFW and our generated fine-grained descriptions is shown in Fig. 4.

Tab. 3 reports the recognition performance of our FineCLIPER compared with other approaches in the zero-shot DFER setting. Not only did we surpass the previous methods when the pretraining data was consistent, but employing our generated fine-grained captions also led to a significant performance improvement. This further demonstrates the effectiveness of the fine-grained description obtained and used by our FineCLIPER, which focuses more on facial changes instead of video scenes (as in MAFW). In other words, fine-grained descriptions play a pivotal role in guiding the model's attention toward detailed aspects of specific facial regions in the zero-shot setting.

## 4.3 Ablation Studies

**Performance w.r.t. middle-level facial features.** We investigate the effectiveness of using the middle-level face semantics obtained by face parsing and landmark detection, and the results are shown in Tab. 4. We have the following observations: 1) By comparing

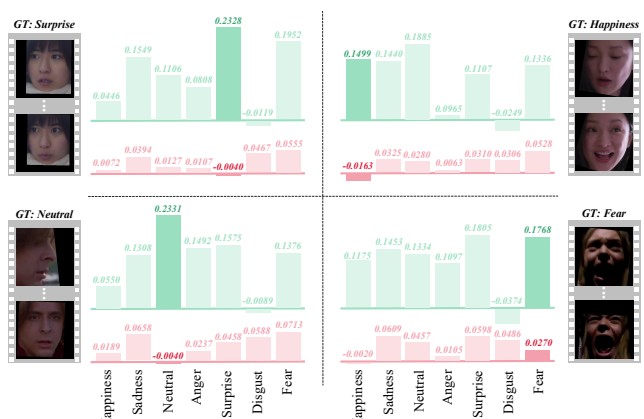

**Figure 5: Visualizations of class-wise cosine similarity values between video and text embeddings in DFEW, where the positive value is in green and the negative one is in red.**

results from rows 1-2, as well as rows 1-3, we find that employing either one kind of the middle-level facial features could improve the performance, justifying the usefulness of middle-level semantic features; 2) Combining both face segmentation and landmarks yields the best results across all datasets, showing their complementary nature and further verifying our choice for using both.

**Performance w.r.t. label augmentation strategies.** Since the DEFR is a classification task, the supervision is originally in the form

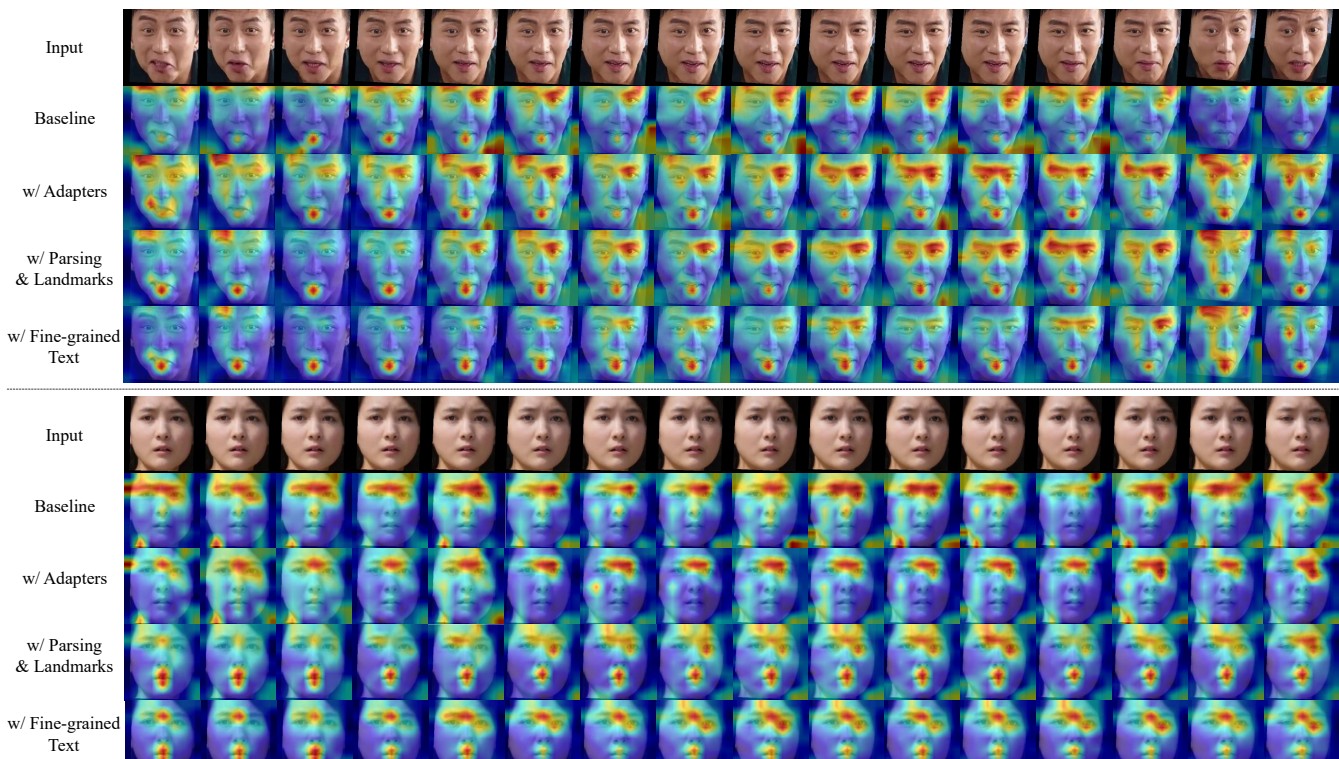

**Figure 6: Attention visualizations for DFEW w.r.t. two ground-truth expression labels—'Happiness' (Top) and 'Surprise' (Bottom).**

of class labels. However, we follow the recent practice of extending the label to semantically textual meaningful supervision and propose a novel idea to construct supervision from both positive and negative aspects. The ablations involving such label augmentation strategy are represented in the first three rows of Tab. 5. When we control other conditions, using our Pos-Neg augmentation achieves the best results across all metrics. Next, to further understand why the Pos-Neg descriptors perform well, we visualize class-wise cosine similarity between video representation and the positive text supervision (colored in green) as well as the negative supervision(colored in red), as shown in Fig. 5. It can be observed that since the positive supervision could sometimes fail to work (as we can see that for some categories the pos-similarity is really low), the existence of its negative counterpart could address such a problem to a certain extent.

**Performance w.r.t. the usage of trainable adapters.** We adopted several lightweight trainable adapters in our FineCLIPER to efficiently adapt the ability of large pre-trained models. The corresponding ablation studies are demonstrated in Tab. 5. We can see that given the same supervision settings (e.g. pose+neg for FineCLIPER), adding small adaptive modules could effectively boost the performance with only limited trainable parameters (e.g. 20M for all adapters in FineCLIPER).

**Effect of each components.** To validate the effectiveness of each component module in our FineCLIPER, we visualize the attention map of the last transformer block, as shown in Fig. 6. Specifically, we sequentially add components from top to bottom, including

adding the Adapterers, using the parsing results and landmarks of faces, as well as using the high-level semantics from the fine-grained descriptions generated by MLLM. We can see that the model's attention is shrinking to more crucial and concentrated face parts w.r.t. to certain categories. For example, it focuses on the mouth, eyes, and eyebrows when identifying *Happiness*, which aligns well with expression recognition using human vision. Such visualization results provide a vivid interpretation to explain the superior recognition performance of FineCLIPER.

## 5 Conclusion

Dynamic Facial Expression Recognition (DFER) is vital for understanding human behavior. However, current methods face challenges due to noisy data, neglect of facial dynamics, and confusing categories. To this end, We propose FineCLIPER, a novel framework with two key innovations: 1) augmenting class labels with textual PN (Positive-Negative) descriptors to differentiate semantic ambiguity based on the CLIP model's cross-modal latent space; 2) employing a hierarchical information mining strategy to mine cues from DFE videos at different semantic levels: *low* (video frame embedding), *middle* (face segmentation masks and landmarks), and *high* (MLLM for detailed descriptions). Additionally, we use Parameter-Efficient Fine-Tuning (PEFT) to adapt all the pre-trained models efficiently. FineCLIPER achieves SOTA performance on various datasets with minimal tunable parameters. Detailed ablations and analysis further verify the effectiveness of each design.

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
