# OpenReview forum: "FineCLIPER: Multi-modal Fine-grained CLIP for Dynamic Facial Expression Recognition with AdaptERs"
_acmmm.org/ACMMM/2024/Conference — MM2024 Poster_

### Official Review · Reviewer_jEhm · 2024-05-05

**Rating:** 5
**Confidence:** 4

**Summary:**

In this work, it proposes a new framework named Multi-modal Fine-grained CLIP for Dynamic Facial Expression Recognition with AdaptERs (FineCLIPER), which combines the following novel designs:1)To better distinguish similar facial expressions, we extend the class labels from positive and negative aspects to textual descriptions and compute cross-modal similarity based on the CLIP model to obtain supervision; 2) FineCLIPER employs a hierarchical approach to efficiently mine useful cues from DFE videos. In addition, the paper uses Parameter Efficient Fine-Tuning (PEFT) to efficiently adapt to all pre-trained models, achieving high efficiency while maintaining excellent performance.

**Strengths:**

The motivation of integrating low level, middle level and high level sematics for expression recognitoin, is clear and the idea is interesting. The results also justify the effectiveness of adding sematics of different levels.

**Limitations:**

About clarity:
(1)	In line 381 of the article, each transformer layer is represented by the superscript i, but below it is represented by j. Is there any difference between the before and after?
(2)	The article mentions that FineCLIPER achieves high efficiency while maintaining excellent performance. However, the article uses multiple models, are they really achieving high efficiency? Can this be illustrated by metrics or graphs?
(3)	Why is a scaling factor s added to Eq. 7 and how is s=0.5 determined, please give appropriate experiments or references.

About experiments:
(1)	The EmoCLIP method in Table 1 is marked as published in FG, but the reference is shown as being from ArXiv.
(2)	What is the performance of the models in Table 4 without the use of the low semantic level, and with the use of the medium or high semantic level alone, respectively?
(3)	In Fig. 5, the video representations of some expressions do not obtain the highest similarity with positive texts (e.g. happiness and fear), please explain why. And elaborate on how the introduction of negative text supervision solves this problem.

**Suitability:**

3

---

### Official Review · Reviewer_wP8p · 2024-05-09

**Rating:** 4
**Confidence:** 4

**Summary:**

The paper proposes multi-modal fine-grained CLIP with adapters for dynamic facial expression recognition.

The novelty of the proposal includes extending positive and negative descriptions and introducing high-semantic level descriptions.

The concerns mainly lie in the assumption of negative descriptors.

**Strengths:**

The proposal is novel. It extends the class label with negative descriptions. It utilizes the MLLM to generate high semantic-level information about facial changes.

The paper conducts extensive experiments. It shows advantages and includes ablation studies in three DFER datasets.

The writing and technical details are clear. The figures and expressions are in good quality. The paper poses the code for reproduction.

**Limitations:**

### 1. The main concern is that the negative descriptors (e.g., a person with an expression of no {label}.) are too absolute.

The facial expression class label is usually ambiguous, e.g., the neutral facial expression might contain fear emotion. Therefore, even if the apparent expression is neutral, we can't say there is no fear. Could you explain it?

### 2. There are some confusing or lack-verification expressions in the paper.
- #L10 ''scarcity of high-quality data". Why do the authors mention it?
- Figure 1. Why do the authors list sample numbers of SFER datasets in DFER?
- #L415 What is the meaning of "P" and "z"?
- #L419 "spatially adapted feature". What is it?
- Figure 2 and #L451. I can't see E_P and E_L share weights from the figure.
- #L469 "rule-based approach refinement". Can you give more details on how to reproduce it?
- #L697 "Zero-Shot DFER methods". I think it is a cross-domain setting since you use the DFER dataset.
- Figure 5. This picture is confusing. For the negative value, is lower better? In the 4th sample, I am confused about Happiness and Fear.
-  Table 5. What do you mean by an "X" of video?

### 3. Typos.
- #L505 "categories" needs a period.
- #L853 "supervision(colored" needs a blank.

**Suitability:**

3

---

### Official Review · Reviewer_nNrv · 2024-05-16

**Rating:** 4
**Confidence:** 2

**Summary:**

This paper proposed a multimodal fine-grained clip for dynamic facial expression recognition.

**Strengths:**

1. The experiments show great performance.
2. The methodology is reasonable.

**Limitations:**

Instead of just using ViT-B/16 as the backbone, have you considered scaling up the model size?

**Suitability:**

3

---

### Official Review · Reviewer_VmG3 · 2024-06-02

**Rating:** 3
**Confidence:** 4

**Summary:**

In this paper, the authors propose a Multi-modal Fine-grained CLIP for Dynamic Facial Expression Recognition with AdaptERs, i.e., FineCLIPER. FineCLIPER adopts a hierarchical manner to enhance visual features, and utilizes the Multi-modal Large Language Model (MLLM) to generate the detailed descriptions of facial changes across frames. Adapters are employed to fine-tune the Encoders. Experiments are conducted on DFEW, FERV39K and MAFEW datasets, showing state-of-the-art performance.

**Strengths:**

This paper employs many popular and novel frameworks and models to complete Dynamic Facial Expression Recognition task, such as Video-LLaVa, FaceXFormer, and Long-CLIP. This work achieves state-of-the-art performance and greatly upgrade the recognizing performance at the category of disgust. This paper is easy to follow. The experiments are enough to validate the efficacy.

**Limitations:**

Though the work keeps up with times, many limitations hinders the understanding of this paper.

1. In introduction section, the authors mentioned current Dynamic Facial Expression Recognition faces two challenges: 1) the ambiguity of semantic labels and 2) subtle movements of local face parts. However, The reason why the multi-modal framework can and how they do to alleviate the mentioned two challenges are not introduced. Sure the detailed operations are given in the following paragraphs, but the overall principle should be given before L113. Lacking the analysis of why the proposed methods can alleviate the corresponding issue makes this work like a simple compound of existing methods. I strongly recommend re-write your introduction section to introduce and issues and clarify the contribution.
2. For contribution 2): employing a hierarchical information mining strategy to mine cues from DFE videos at different semantic levels. How the proposed method solve one certain issue should be clarified, otherwise, this framework is considered as trivial and incremental design.
3. For contribution 3): Parameter-Efficient Fine-Tuning has widely used in existing works. It seems a basic trick for fine-tuning large model.
4. The paper has some confused symbols. L283: Does the meaning of i in \textbf{v_i} is the same as class i? Please clarify it. If it is, please remove the bold style for "i". L 288: What is the formula to calculate the similarity between the video feature and textual supervision? L551 formula(11): Missing the meaning of B, H in formula (11).

**Suitability:**

3

---

### Meta-Review · Area_Chair_UAZV · 2024-07-02

**Recommendation:** Accept (Poster)
**Confidence:** 4

**Metareview:**

This paper proposes multi-modal fine-grained CLIP with adapters for facial expression recognition.

Most reviewers tend to accept this paper. The authors should polish this paper in the final version by considering the comments of Reviewer VmG3. It is suggested to make the paper clearer including discussing the motivation why the designed method can address the stated issues.